

# Biochemical and molecular characterization of N66 from the shell of *Pinctada mazatlanica*

Crisalejandra Rivera-Perez[1], Catalina Magallanes-Dominguez[2],
Rosa Virginia Dominguez-Beltran[3], Josafat Jehu Ojeda-Ramirez
de Areyano[2] and Norma Y. Hernandez-Saavedra[2]

[1] Department of Fisheries Ecology, CONACyT-Centro de Investigaciones Biologicas del Noroeste (CIBNOR), La Paz, Baja California Sur, Mexico
[2] Department of Fisheries Ecology, Molecular Genetics Laboratory, Centro de Investigaciones Biologicas del Noroeste (CIBNOR), La Paz, Baja California Sur, Mexico
[3] Tecnológico Nacional de México, La Paz, Baja California Sur, Mexico

## ABSTRACT

Mollusk shell mineralization is a tightly controlled process made by shell matrix proteins (SMPs). However, the study of SMPs has been limited to a few model species. In this study, the N66 mRNA of the pearl oyster *Pinctada mazatlanica* was cloned and functionally characterized. The full sequence of the N66 mRNA comprises 1,766 base pairs, and encodes one N66 protein. A sequence analysis revealed that N66 contained two carbonic anhydrase (CA) domains, a NG domain and several glycosylation sites. The sequence showed similarity to the CA VII but also with its homolog protein nacrein. The native N66 protein was isolated from the shell and identified by mass spectrometry, the peptide sequence matched to the nucleotide sequence obtained. Native N66 is a glycoprotein with a molecular mass of 60–66 kDa which displays CA activity and calcium carbonate precipitation ability in presence of different salts. Also, a recombinant form of N66 was produced in *Escherichia coli*, and functionally characterized. The recombinant N66 displayed higher CA activity and crystallization capability than the native N66, suggesting that the lack of posttranslational modifications in the recombinant N66 might modulate its activity.

## INTRODUCTION

Mollusks are known for their ability to build shells of different size, forms and structures. Mollusk shells are mainly composed of calcium carbonate (aragonite, calcite, and nacre), and synthesized by shell matrix proteins (SMPs) (*Addadi et al., 2006*). SMPs are synthesized in the mantle and secreted to the extrapalleal space, between the shell and the mantle, where they interact with bicarbonate ions ($HCO^{3-}$), calcium ($Ca^{2+}$), polysaccharides, and metal traces ($Mg^{2+}$) (*Xie et al., 2016*; *Liu et al., 2011*), leading to aragonite (nacre, crossed-lamellar, and complex crossed-lamellar) and calcite microstructures (prismatic and foliated) (*Zhang & Zhang, 2006*). The nacre layer of the shell consists of organized crystal aragonite tablets forming a brick and mortar

Corresponding author
Norma Y. Hernandez-Saavedra,
nhernan04@cibnor.mx

construction (*Mutvei, 1980*) which displays outstanding biomechanical properties, such as toughness, elasticity, light weight or softness (*Checa, Macías-Sánchez & Ramírez-Rico, 2016*).

Although the aragonite assembly process is not well understood, SMPs involved in aragonite crystal growth have been identified in several marine mollusks (*Zhang & Zhang, 2006*) through transcriptomic and proteomic analysis (*Miyamoto et al., 2013*; *Mann, Edsinger-Gonzalez & Mann, 2012*). Nacrein and N66 proteins play a significant role in the biomineralization process, they are soluble enzymes involved in the crystallization of calcium carbonate and take part in the nacreous layer formation (*Miyamoto et al., 1996*). Nacrein and N66 are metalloenzymes with one Gly-Xaa-Asn repeating domain (Xaa: Asp, Asn, or Glu, NG-repeat domain) at the C-terminal end (*Song et al., 2014*; *Mann, Edsinger-Gonzalez & Mann, 2012*; *Norizuki & Samata, 2008*; *Kono, Hayashi & Samata, 2000*; *Miyamoto, Yano & Miyashita, 2003*; *Miyamoto et al., 1996*) and two carbonic anhydrase (CA) domains which belong to the CA superfamily. All nacrein and N66 proteins described in mollusks belong to the α-CA family, class II and VII, respectively (*Mann, Edsinger-Gonzalez & Mann, 2012*; *Leggat et al., 2005*; *Kono, Hayashi & Samata, 2000*; *Miyamoto et al., 1996*). These proteins are known to contain a metal ion ($Zn^{2+}$) coordinated by three His residues at the active site (*Ozensoy, Capasso & Supuran, 2016*) and catalyze the reversible hydration of carbon dioxide to bicarbonate (*Miyamoto et al., 1996*).

Nacrein and N66 are conserved in bivalves and gastropods and have a molecular weight ranging from 50 to 66 kDa (*Song et al., 2014*; *Mann, Edsinger-Gonzalez & Mann, 2012*; *Leggat et al., 2005*; *Miyamoto, Yano & Miyashita, 2003*; *Kono, Hayashi & Samata, 2000*; *Miyamoto et al., 1996*). Previous studies have recognized that nacrein and N66 are glycoproteins (*Leggat et al., 2005*; *Kono, Hayashi & Samata, 2000*), where nacrein possess N-glycan containing sulfite and sialic acid at its terminus (*Takakura et al., 2008*), furthermore, they both have calcium binding ability (*Miyamoto et al., 1996*). Despite their similar biochemical properties, N66 has almost twice length of NG repeat domain than nacrein (*Miyamoto, Yano & Miyashita, 2003*; *Kono, Hayashi & Samata, 2000*; *Miyamoto et al., 1996*). This domain has been proved to function as a negative regulator in the shell formation of nacrein (*Miyamoto, Miyoshi & Kohno, 2005*), and as a positive regulator, both as a $Ca^{2+}$ concentrator and an enzyme required for production of carbonate ions (*Norizuki & Samata, 2008*). Longer repeats of the NG domain have been related to a stronger reaction between proteins with $Ca^{2+}$ ions, matrix components and crystals, resulting in an improved calcification capability (*Samata et al., 1999*).

The nacrein gene has been described in the bivalve *Pinctada maxima* (*Kono, Hayashi & Samata, 2000*) and the gastropod *Turbo marmoratus* (*Miyamoto, Yano & Miyashita, 2003*), while the N66 gene has only been described in *Pinctada maxima* (*Kono, Hayashi & Samata, 2000*). Differential expression of SMP mRNAs in the mantle of mollusks has been related to their crystal layer formation. Nacrein mRNA is expressed in the mantle edge and pallial (*Miyamoto, Miyoshi & Kohno, 2005*; *Miyamoto et al., 1996*), while N66 mRNA is expressed in the dorsal region of the mantle and the mantle edge (*Kono, Hayashi & Samata, 2000*), the former has been associated to function of the nacreous layer and the latter to the prismatic layer formation (*Sudo et al., 1997*).

To gain insights into the biomineralization process in mollusk and to increase the knowledge of the nacre formation, a comprehensive study of the proteins involved in nacre deposition is required. Pearl oyster, *Pinctada mazatlanica* is a good model to study biomineralization, since includes two layers, an inner nacreous layer, and an outer prismatic calcite layer. In this study, a N66 from the shell was isolated and its coding sequence was obtained from the mantle of the pearl oyster *Pinctada mazatlanica* and overexpressed on a prokaryotic system, the resulting native and recombinant proteins were functionally characterized.

## MATERIAL AND METHODS

### Biological material

Three adult female oysters were provided by Perlas del Cortez S. de R.L. MI. located at Bahia de La Paz B.C.S. Organisms were transported to the Molecular Genetics Laboratory at CIBNOR, and mantle tissue was dissected and stored at −80 °C until used.

### Molecular characterization of N66 from mantle tissue

#### *Total RNA extraction and RNA reverse transcription*

Total RNA was extracted with TRIzol® (Invitrogen, Carlsbad, CA, U.S.A.) according to the manufacturer's instructions. Samples were homogenized using a glass pestle; then, two consecutive extractions of each sample were made. RNA purity and concentration were determined by spectrophotometry using a NanoDrop ND-2000 (Thermo Scientific, Waltham, MA, U.S.A.) at 260/280 and 260/230 nm absorbance ratios (range, 1.9–2.0). The RNA integrity was assessed on a 1% (w/v) agarose-synergel gel. To ensure complete DNA absence, a direct PCR was performed using one μL (50 ng/μL) of each RNA preparation with 28S ribosomal specific primers as a non-amplified control. After that, one μg of total RNA was used from each verified RNA sample for cDNA synthesis using the cloned AMV First-Strand cDNA Synthesis Reaction (Invitrogen, Carlsbad, CA, U.S.A.) and oligo-dT primer, afterwards cDNAs were stored at −80 °C until use. Control reactions were performed without template or non-reverse transcribed RNA to determine the presence of DNA.

#### *Amplification of the N66 coding sequence from mantle*

A search for mollusk sequences encoding N66 available at GenBank yielded three homologous sequences, N66 *Pinctada maxima* (GenBank: AB032613), N44 *Pinctada maxima* (GenBank: FJ913472), and SP-S *Mizuhopecten yessoensis* (GenBank: AB185328). Based on these sequences, specific primers were designed to amplify the coding sequence of N66 (Table 1). PCR amplification of cDNA fragments encoding N66 was done in a final volume of 12.5 μL containing 10 pmol of each forward and reverse primer, 50 ng cDNA, 10 nmol dNTPs mixture, 1.25 μL 10X Taq Buffer, and 1 U Taq DNA polymerase (11615010A; Invitrogen, Carlsbad, CA, USA). PCR amplification were carried out for 5 min at 94 °C, followed by five cycles of 1 min at 94 °C, 1 min at 40 °C, and 2 min at 72 °C, followed of 35 cycles of 1 min at 94 °C, 1 min at 50 °C, and 2 min at 72 °C. In the last cycle, the extension step at 72 °C lasted 10 min. PCR products were analyzed on 1% (w/v) agarose-synergel gels and visualized under UV light after staining with UView loading dye

**Table 1 Primer sequences and expected amplicon size for N66 amplification.**

| Primers | Primer sequence (5′-3′) | Amplicon size (pb) |
|---|---|---|
| N66_1F | ATGTGGAGAATGACGACGCTT | 425 |
| N66_425R | TGTGAATATGGAGATTGCGG | |
| N66_1F | ATGTGGAGAATGACGACGCTT | 680 |
| N66_680R | CCTTTAATATGCGCTTACATTC | |
| N66_680Fw | GAATGTAAGCGCATATTAAAGG | 962 |
| N66_1642Rv | GGGACGTCTGGTTCCATAC | |
| N66_1F | ATGTGGAGAATGACGACGCTT | 1,766 |
| N66_1766R | CTCATATAAAGTTTTTGTACACAGG | |
| N66_HindF | GTG**AAGCTT**TGCCTCCATGCACAGGCATG | 1,642 |
| N66_HindR | CAC**AAGCTT**GGGACGTCTTGTTTCCATAC | |

(166-5112; Bio-Rad, Hercules, CA, USA). PCR products were cloned using the TOPO-TA cloning kit (K4500-01; Invitrogen, Carlsbad, CA, USA) and both strands were sequenced. The obtained sequence was submitted to the National Center for Biotechnology Information (NCBI) database for BLAST searching and other informatic analysis.

### In silico analyses

Identification of the open reading frame was performed using the ORF finder software (https://www.ncbi.nlm.nih.gov/orffinder/) and sequence alignment with homologous sequences was performed using CLUSTAL Omega. Identification of putative protein motifs was performed using the MotifScan (Pfam HMMs global models database) available at the Swiss Institute of Bioinformatic (http://myhits.isb-sib.ch/cgi-bin/motif_scan). Identification of signal peptide was achieved by using SignalP 4.1 Server (*Petersen et al., 2011*). Theoretical molecular weight, isoelectric point (pI), and amino acid composition of the protein were calculated using the ProtParam software from Expert Protein Analysis System (ExPASy; https://www.expasy.org/). Putative glycosylation and phosphorylation sites of N66 were determined using NetOGly4.0 (*Steentoft et al., 2013*), NetNGlyc 1.0 (http://www.cbs.dtu.dk/services/NetNGlyc/), and NetPhos2.0 (*Blom, Gammeltoft & Brunak, 1999*).

### Vector construction and recombinant protein expression

The coding region of the N66 cDNA, without the signal peptide, was synthesized by PCR with two primers (N66_HindF and N66_HindR) to introduce the restriction enzyme sites (HindIII, AAGCTT) at the 5′ and 3′ ends (Table 1). The PCR product was directly digested with HindIII (NEB, New England, Ipswich, MA, USA) and isolated from gel electrophoresis, purified and ligated into a polyhistidine fusion protein expression vector pTrcHis C (Invitrogen Inc., Carlsbad, CA, USA) that was linearized with the same restriction enzyme, yielding pTrcHis C/N66. *Escherichia coli* TOP10 cells were transformed with the construct and cultured in six mL of LB medium containing 100 μg/mL ampicillin at 37 °C with shaking at 150 rpm overnight. Five mL of the culture medium were transferred to a flask containing 245 mL of the same medium and further incubated. When $OD_{600}$ of the bacterial cells reached ~0.7, isopropyl β-D-thiogalactosidase was

added to a final concentration of one mM and incubated at 37 °C with constant shaking at 150 rpm for 4 h. The bacterial cells were harvested by centrifugation (7,500 rpm, 25 min at room temperature) and stored at −20 °C. The frozen bacterial pellet was thawed, dispersed in ice-cold Tris–HCl buffer (50 mM, pH 7.4) and disrupted with glass beads for 5 min of vigorous vortex. The bacterial lysate was centrifuged (12,000×$g$ rpm, 15 min, 4 °C), and the supernatant was recovered. Recombinant His-tagged proteins were purified from the supernatant using a Sepharose-Ni column (GE Healthcare, Chicago, IL, USA).

## Native N66 purification and characterization
### Shell matrix extraction
The organic matrix of prismatic and nacreous layers was crushed into a fine powder. The powdered matrix (20 g) was suspended in 100 mL of cold acetic acid (4 °C, 10% v/v) and incubated for 24 h with continuous stirring. Soluble (ASM) and insoluble (AIM) matrix fractions were separated by centrifugation at 14,000×$g$ rpm for 20 min. The AIM was rinsed with distilled water and lyophilized. After that, the ASM was dialyzed 24 h against cold acetic acid (4 °C, 1% v/v), afterward the ASM was dialyzed another 24 h against distilled water. The dialyzed ASM was concentrated by lyophilization. Protein concentration was determined with the Lowry method (*Lowry et al., 1951*).

### N66 purification by affinity chromatography
Native and recombinant N66 were isolated by affinity chromatography using a Sepharose-Ni column (GE Healthcare, Chicago, IL, USA; 1.45 × 5.0 cm). The protein sample (one mL) was loaded onto the column, previously equilibrated with 50 mM Tris–HCl pH 7.4, 300 mM NaCl buffer, mixed and incubated for 1 h at 25 °C. The unbound enzyme was washed with five bed volumes of 50 mM Tris–HCl pH 7.5, 300 mM NaCl buffer. The bound enzyme was eluted by washing the column with elution buffer containing 50 mM Tris–HCl pH 7.5, 300 mM NaCl, 300 mM Imidazole. Fractions of one mL were collected from the elution and evaluated by electrophoresis. Fractions containing the N66 protein were pooled and concentrated through centrifugal filters (AMICON Ultra-10; Millipore, Burlington, MA, USA) at 4,000×$g$ for 20 min at 25 °C to a final volume of 500 µL. The concentrated sample was desalted using a PD-10 desalting column (Amershan Pharmacia, Little Chalfont, UK). Glycerol was added to the enzyme solution (final concentration 50%), and the sample was stored at −20 °C until use. Protein content was quantified with the Lowry method (*Lowry et al., 1951*) and sodium dodecyl sulfate polyacrylamide gel electrophoresis (SDS-PAGE) analysis of the purified protein was performed.

### SDS-PAGE and staining
Native and recombinant N66 proteins were analyzed under reducing conditions using 10% SDS-PAGE (Mini-PROTEAN; Bio-Rad, Hercules, CA, USA), as described by *Laemmli (1970)* and stained with silver nitrate (*Merril & Washart, 1998*) in 10% acrylamide gels. Samples were diluted (1:4) with sample buffer (0.125M Tris–HCl, 2% SDS, 20% v/v glycerol, 0.04% bromophenol blue, 5% β-mercaptoethanol at pH 6.8) and heated for 10 min at 100 °C.

The electrophoresis was performed at 80V constant voltage. To detect protein bands, gels were stained with 0.1% Coomassie Brilliant Blue R-250 in 7.5% acetic acid and 5% methanol at room temperature and distained in the 10% acetic acid and 40% methanol.

Glycosylation was also analyzed by electrophoresis under reducing conditions (*Thornton, Carlstedt & Sheehan, 1994*). The gel was first incubated in a fixing solution (5% TCA) for 5 min, then incubated in an oxidant solution (0.8% periodic acid in 0.3% sodium acetate) and rinsed with water. Then, the gel was incubated in Schiff's reagent for 10 min in the dark. Finally, the gel was washed three times with 3% nitric acid for 3 min. The gel was stored in 7.5% acetic acid, 5% v/v methanol. Ovoalbumin (Sigma-Aldrich, St. Louis, MO) was used as positive control.

### Mass spectrometry analyses

To identify the mass spectrometric profile of the native N66, 20 µg of protein were separated by 10% SDS-PAGE. After electrophoresis, the gel was stained with Coomassie Blue R-250 and bands were manually cut from the gel. After tryptic digestion, the resulting peptides were subjected to Liquid Chromatography-Mass Spectrometry (LC-MS). The LC-MS was performed at the Proteomic Unit Facility at the Biotechnology Institute, in Cuernavaca, Mexico. The resulting data were compared with the proteomic database at GenBank and with the deduced amino acid sequence from cDNA of N66 from *Pinctada mazatlanica*.

### Carbonic anhydrase assay

Carbonic anhydrase activity from native and recombinant N66 was assayed qualitatively and quantitatively. In the qualitative assay, 1% agarose plates were prepared with borate buffer (0.05M boric acid, 0.025M NaCl). Samples (10 µL with 0.2 mg protein/mL) were loaded on the agar plate. After sample diffusion 0.1% bromocresol purple (dissolved in 0.1M Tris–HCl pH 8.5) was added and incubated for 10 min at room temperature. Afterwards, the solution was discarded and the agar plate was washed with distilled water. Finally, saturated $CO_2$ water was added as substrate; the production of hydrogen ions during the $CO_2$ hydration reaction lowers the pH of the solution until the formation of yellow halo zones which indicate CA activity while dark zones indicate lack of activity (*Manchenko, 2002*). Bovine CA (Sigma-Aldrich, St. Louis, MO) was used as positive control and it was boiled to function as a negative control.

The quantitative assay for CA activity is described in detail by *Wilbur & Anderson (1948)*. CA activity was measured by the decrease of pH resulting from the hydration of $CO_2$ to $HCO_3^-$ and $H^+$ after the addition of the substrate. All experiments were performed at 4 °C. To run the assay, five mL of 0.012M Tris–HCl buffer (pH 8.3) were transferred to a small beaker and one mL of the homogenate diluted in 0.012M Tris–HCl buffer was added to obtain 20 µg of protein. The mixture was constantly stirred with a magnetic stirring bar. Afterwards, four mL of cold saturated $CO_2$ solution were added, and the decrease in pH was recorded by a pH probe. CA activity was calculated as $(t_0-t)/t$, where $t_0$ is the time needed for the non-catalyzed reaction and $t$ is the time for the catalyzed

reaction to obtain a pH decrease from 8.0 to 6.0. Units of enzyme activity (U) were normalized to the total protein content.

### Crystallization assay with native and recombinant N66 proteins

Three crystallization solutions were prepared, two induces calcite (calcitic crystallization solution) and one induces aragonite (aragonite crystallization solution). Calcitic crystallization solutions (40 mM CaCl$_2$, pH 8.2, 100 mM NaHCO$_3$, and 100 mM CaCO$_3$) and aragonite crystallization solution (40 mM MgCl$_2$, pH 8.2, 100 mM NaHCO$_3$) were prepared following the protocol of *Weiss et al. (2000)* and *Hillner et al. (1992)*, respectively. Native and recombinant N66 (10 µL of 0.2 mg/mL) were added to 50 µL of the saturated solution. As a control, buffer solution without protein was used. Each experiment was made in triplicates and incubated at 4 °C for 21 days in a cover slide situated at the bottom of a six-hole microplate which was sealed with parafilm to avoid contamination. The morphology of the crystals produced were analyzed by scanning electron microscopy (SEM) at the Electronic Microscopy Laboratory at CIBNOR.

### Statistical analysis

Statistical analysis was performed using the GraphPad Prism Software. The results are expressed as means ± standard deviations. Significant differences ($P < 0.05$) were determined by ANOVA followed by a pair-wise comparison of means (Tukey's test).

## RESULTS

### Sequence analysis of cDNA from mantle of *Pinctada mazatlanica* encoding a N66

Nucleotides and deduced amino acid sequence of N66 from the mantle of *Pinctada mazatlanica* are shown in Fig. 1. The open reading frame of N66 obtained from the mantle cDNA was 1,766 bp long (GenBank: MH473230) and encoded a 576 amino acid protein with a calculated molecular mass of 63.68 kDa before any post-translational modification (PTM) and an pI of 7.38. Removal of the signal peptide sequence (residues 1–22) resulted in a theoretical molecular weight of 61.12 kDa and a theoretical pI of 7.21. It possesses high proportions of N (19.6%) and G (15.8%), which accounted for 35.4% of the total amino acid residues (Table 2). Analysis of the N66 amino acid sequence showed homology with the CA family, exhibiting 34% identity with human CAVII (GenBank: NP_005173), with high homology to amino acid residues involved in catalysis ($Q_{125}$, $N_{154}$, and $E_{168}$) and three histidine residues essential for the zinc cofactor binding ($H_{156}$, $H_{158}$, and $H_{181}$) (Fig. 1).

The N66 protein sequence possesses in its modular structure a signal peptide (1–22 aa), two CA domains ($G_{69}$-$G_{221}$ and $D_{490}$-$K_{554}$), and a NG rich sequence ($N_{245}$-$G_{437}$) (Fig. S1), sharing 82.82% identity to the N66 from *Pinctada maxima*, ~92% to nacreins (B2, B3, and B4) from *Pinctada margaritifera*, 72.62% to nacrein from *Pinctada fucata* and 78.70% to N45 from *Pinctada maxima* (Fig. 2). All the protein sequences contained the same protein structure, however the length of the CA and NG domain varied among the proteins.

Further bioinformatic analyses of the N66 sequence suggests that some serine and threonine residues in N66 may either be phosphorylated (36 putative sites) or N- and
```
   1  tatgtggagaatgacgacgcttcttcacttgactgctctgcttgttctgattccattatgt
   1  M  W  R  M  T  T  L  L  H  L  T  A  L  L  V  L  I  P  L  C
  62  cattgtgcctccatgcacaggcatgaccattatatggacatggatcaaacctaccgtgat
  21  H  C  A  S  M  H  R  H  D  H  Y  M  D  M  D  Q  T  Y  R  D
 121  cgatggggaaactgtcattattcaggggaagtagtgtgacgccgggtttagctacaat
  41  R  W  G  N  C  H  Y  S  G  G  S  S  C  D  A  G  F  S  Y  N
 181  agggaacaaaatgaggaacaatgccacgggccgtatgactggcacactatatctagttgc
  61  R  E  Q  N  E  E  Q  C  H  G  P  Y  D  W  H  T  I  S  S  C
 241  tttaaggcatgtggaagtaaagagagacaatcaccaatcaacatttggtcacatagagcc
  81  F  K  A  C  G  S  K  E  R  Q  S  P  I  N  I  W  S  H  R  A
 301  cttttccgaaaactgccaagactgaaattcaagccacatatgaaatcattggatacgaaa
 101  L  F  R  K  L  P  R  L  K  F  K  P  H  M  K  S  L  D  T  K
 361  gtgtcaaatcaccaaaatcatgcccctgaattcgattctgaggacgaaaaacttcatgtt
 121  V  S  N  H  Q  N  H  A  P  E  F  D  S  E  D  E  K  L  H  V
 421  aaactgaagaaacttgttgatggacattataaattccgcaatctccatattcacattggc
 141  K  L  K  K  L  V  D  G  H  Y  K  F  R  N  L  H  I  H  I  G
 481  aaaagtagacgaaagggctccgaacacagcgttgacagacattttacacctatggagct
 161  K  S  R  R  K  G  S  E  H  S  V  D  R  H  F  T  P  M  E  A
 541  catttagtgttccatcatgatgagaaaaaggaaatcaaacctcctaggattccgttagga
 181  H  L  V  F  H  H  D  E  K  K  E  I  K  P  P  R  I  P  L  G
 601  agaaatttcagtggtattaatgaatttgttgtcgttggggttttttctagaggatggtgat
 201  R  N  F  S  G  I  N  E  F  V  V  V  G  V  F  L  E  D  G
 661  gaaggatacggtgatgaaccggacgactatgaatgtaagcgcatattaaagggtcattac
 221  E  G  Y  G  D  E  P  D  D  Y  E  C  K  R  I  L  K  G  H  Y
 721  gatcattgcgacaacaatggagacaacggttacaactgtgataacggcaacaatgaaaac
 241  D  H  C  D  N  N  G  D  N  G  Y  N  C  D  N  G  N  N  E  N
 781  aacggaaacaatggtaatggtaacaacggctataacggtaacaacggttataacggtaat
 261  N  G  N  N  G  N  G  N  N  G  Y  N  G  N  N  G  Y  N  G  N
 841  aacggtgacaatggcaacagtggaaacaatggtaatggtaacaacggttataacggtgac
 281  N  G  D  N  G  N  S  G  N  N  G  N  G  N  N  G  Y  N  G  D
 901  aatggcaacagcggaaacaatggtaatggtaacaacggtaataacggtaataacggtggc
 301  N  G  N  S  G  N  N  G  N  G  N  N  G  N  N  G  N  N  G  G
 961  aacggaaacaacagaaacaatggtaatggtaacaacggttataacggtaataacggtgat
 321  N  G  N  N  R  N  N  G  N  G  N  N  G  Y  N  G  N  N  G  D
1021  aatggcaacaacggaaacaatggtaatggtaacaacggaaataatggtaatggtaacaac
 341  N  G  N  N  G  N  N  G  N  G  N  N  G  N  N  G  N  G  N  N
1081  ggaaataacggtggcaatggcaacaacggaaacaatggtaatggtaacaacggaaataat
 361  G  N  N  G  G  N  G  N  N  G  N  N  G  N  G  N  N  G  N  N
1141  ggtaatggtaacaacgggaataacggtggcaatggcaacaatggtaatggtaacaatgga
 381  G  N  G  N  N  G  N  N  G  G  N  G  N  N  G  N  N  G  N  N
1201  agtaatggtaacggtgactacggtagtaatggtaacaatggtggaaacgggaacaatggt
 401  S  N  G  N  G  D  Y  G  S  N  G  N  N  G  G  N  G  N  N  G
1261  aataacggtgataacggtaatggcgacaatggttataacggtgataatggtaacagtgac
 421  N  N  G  D  N  G  N  G  D  N  G  Y  N  G  D  N  G  N  S  D
1321  gggcgactcagacgttgggacttggaaaatgtccgccgcattcataccgagcgatatcac
 441  G  R  L  R  R  W  D  L  E  N  V  R  R  I  H  T  E  R  Y  H
1381  atcagcggaagatgtattgtcaaaaaagcaaaacgcctcagcaggattctcgaatgcgca
 461  I  S  G  R  C  I  V  K  K  A  K  R  L  S  R  I  L  E  C  A
1441  tatagacacaaaaaagtcagagaattcaaaaggaatggaacacaaaagtcttgatgtt
 481  Y  R  H  K  K  V  R  E  F  K  R  N  G  E  H  K  S  L  D  V
1501  gaaattacaccggaaatggttctaccgccaataaaatacagacaatactatacctatgaa
 501  E  I  T  P  E  M  V  L  P  P  I  K  Y  R  Q  Y  Y  T  Y  E
1561  ggatctttgacaacccctccttgcacagagagcgtccgttgggttgtagaaaaatgccac
 521  G  S  L  T  T  P  P  C  T  E  S  V  R  W  V  V  E  K  C  H
1621  gtgcaagtatccagaagggtgcttgatgcattgcggaaggtcgaaggatatgatgatggt
 541  V  Q  V  S  R  R  V  L  D  A  L  R  K  V  E  G  Y  D  D  G
1681  accacgtttgagcaagtatggaaacaagacgtcccacacagagaaacataaaacctgtac
 561  T  T  F  E  Q  V  W  K  Q  D  V  P  H  R  E  T  -
1741  ctgtgtacaaaaactttatatgaga
```

**Figure 1 Nucleotide and deduced amino acid sequence in *P. mazatlanica* N66 cDNA (GenBank: MH473230).** The putative signal peptide is underlined. Conserved anhydrase carbonic domains are highlighted by shaded boxes. Triangles indicate amino acid residues involved in catalysis. Putative phosphorylation sites are shown in bold letters, and amino acids representing N- and O-glycosylation are circle and boxed respectively. The putative polyadenylation signal (AATAAA) is indicated in italics. The peptide sequences obtained by mass spectrometry analyses are boxed with dotted line.

**Table 2  Amino acid composition of N66 from mantle of *P. mazatlanica*.**

| Amino acid residue | Residues | Mol% |
|---|---|---|
| Ala | 10 | 1.7% |
| Arg | 35 | 6.1% |
| Asn | 113 | 9.6% |
| Asp | 35 | 6.1% |
| Cys | 14 | 2.4% |
| Gln | 9 | 1.6% |
| Glu | 30 | 5.2% |
| Gly | 91 | 5.8% |
| His | 29 | 5.0% |
| Ile | 6 | 2.8% |
| Leu | 28 | 4.9% |
| Lys | 30 | 5.2% |
| Met | 8 | 1.4% |
| Phe | 13 | 2.3% |
| Pro | 17 | 3.0% |
| Ser | 28 | 4.9% |
| Thr | 16 | 2.8% |
| Trp | 7 | 1.2% |
| Tyr | 23 | 4.0% |
| Val | 24 | 4.2% |
| Mr | 63.68 | |
| pI | 7.38 | |

O-glycosylated (16 putative sites), as indicated in Fig. 1, which are common features related to proteins involved in biomineralization (*Nudelman et al., 2007*). Finally, most of the O-glycosylation and phosphorylation sites were found in the CA domain.

## N66 protein characterization

### Native N66 from Pinctada mazatlanica shells

The organic matrix of the shell from *Pinctada mazatlanica* was separated by SDS-PAGE revealing a wide range of abundant molecular weight proteins in the ASM fraction (Fig. 3A). The ASM fraction was used to isolate the N66 by affinity chromatography using a Sepharose-Ni matrix (Fig. S2) since the active site of the CA domain contains three histidines involved in metal ion binding which has been shown to be conserved in N66 from *Pinctada mazatlanica* (this study) and *Pinctada maxima* (*Kono, Hayashi & Samata, 2000*). The fractions containing the eluted N66 (Fig. S2) were pooled and desalted. Two bands in SDS-PAGE and silver stained gels were observed (Fig. 3B) with molecular weights of ~60 and 64 kDa. A flow sheet of the purification process for N66 is presented in Table 3. The enzyme was enriched 3.39-fold and the yield was 56.52%. Electrophoretic analysis of N66 with PAS stain, demonstrated that N66 is glycosylated (Fig. 3B); however, the type of carbohydrate associated was not analyzed. The results of LC-MS of the bands matched the deduced amino acid sequence of the N66 of the mantle of *Pinctada*

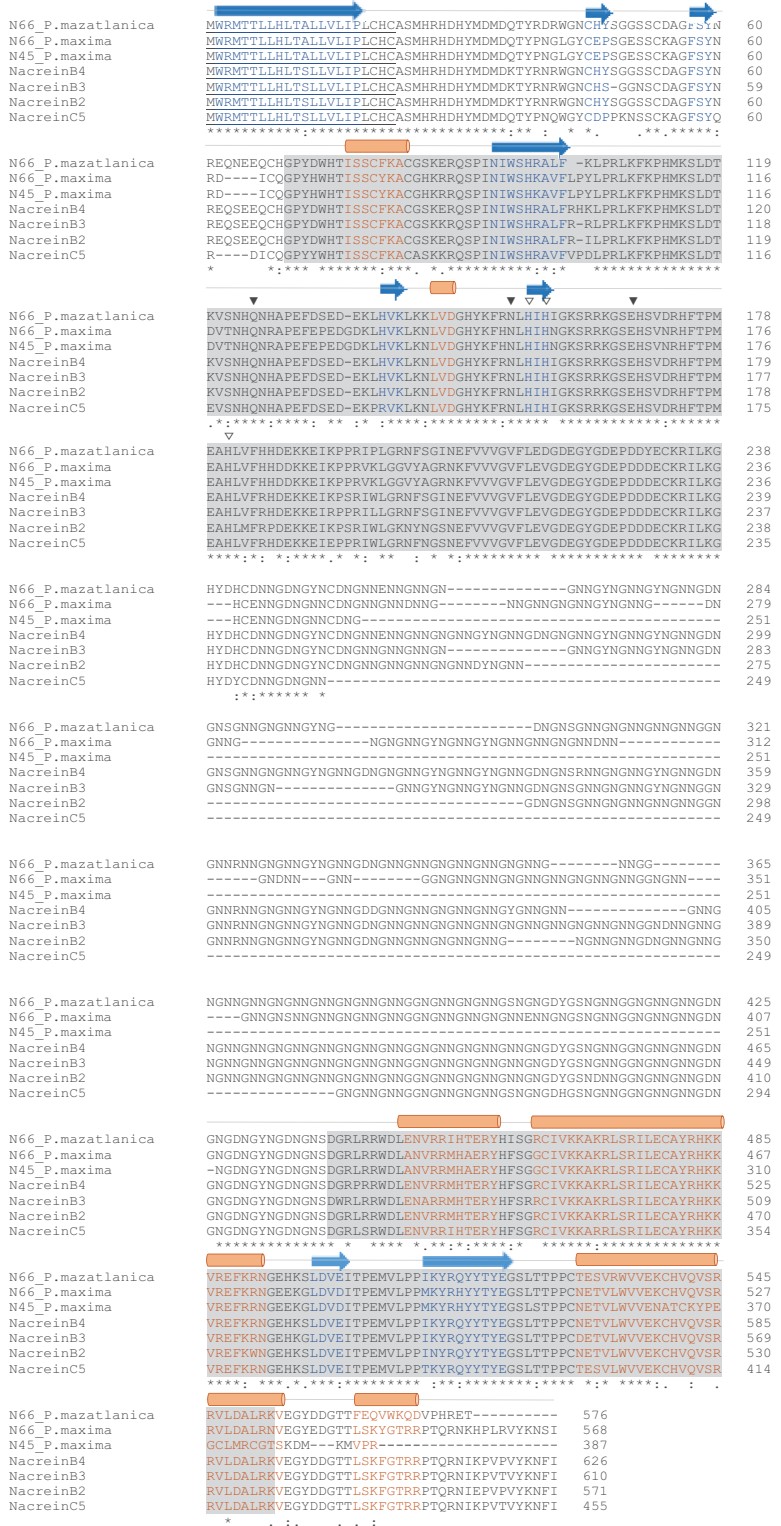

**Figure 2 Amino acid sequence alignment of N66 and related shell matrix proteins.** Accession numbers: *P. mazatlanica* N66 (this study, GenBank: MH473230), *P. maxima* N66 (GenBank: BAA90540.1), *P. fucata* Nacrein (GenBank: BAA119401.1), *P. margaritifera* nacrein B2 (GenBank: ADY69618.1), B3 (GenBank: AEC03971.1), B4 (GenBank: AEC03972.1) and C5 (GenBank: AEC03973.1), and *P. maxima*
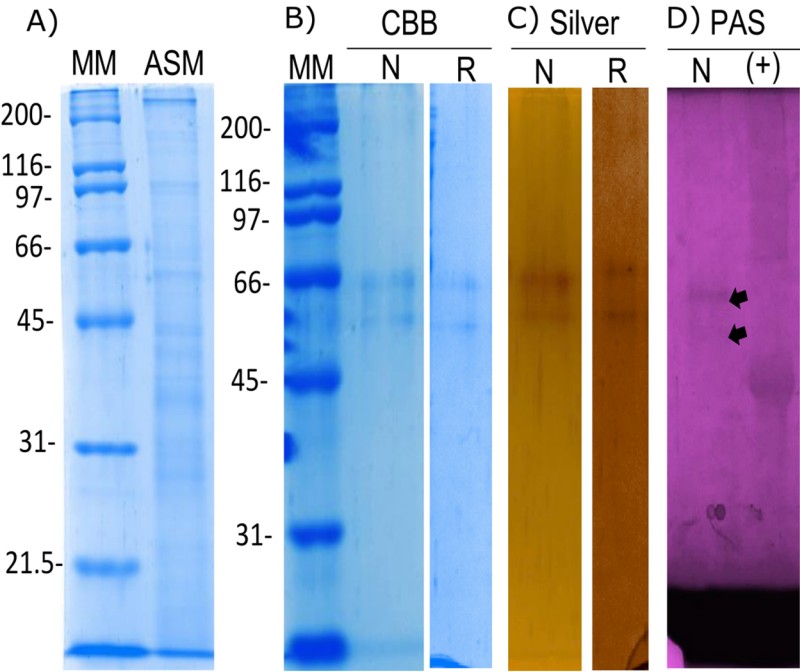

**Figure 2** (continued)
N45 (GenBank: ACT55367.1). Signal peptide is underlined; conserved carbonic anhydrase domains are highlighted by shaded boxes. Triangles indicate amino acid residues involved in catalysis, open triangles indicates zinc binding histidines. Alpha helix (cylinders) and beta sheets (arrows) are shown along the amino acid sequence. Below the protein sequences is a key denoting conserved sequence (*), conservative (:) and semi-conservative (.) modifications.

**Figure 3** **Biochemical characterization of the native and recombinant N66 from *P. mazatlanica*.** Analysis by 10% SDS-PAGE. (A) Acetic soluble matrix protein extract; (B) CBB, Coomassie Brilliant Blue; (C) Silver nitrate stain; and (D) PAS, Periodic acid/Schiff staining. MM, molecular marker; N, Native protein; R, Recombinant protein and Ovoalbumin as positive control (+) for PAS.

**Table 3** **Summary of the native N66 purification from the shell of *P. mazatlanica*.**

| Step | Protein (mg/mL) | Total protein (mg) | Total activity (U) | Specific activity (U/mg) | Purification (fold) | Yield (%) |
|---|---|---|---|---|---|---|
| Acetic acid extraction-dialysis | 19.20 | 19.20 | 990.72 | 51.60 | 1 | 100 |
| Sepharose-Ni | 0.32 | 3.20 | 560.00 | 175.0 | 3.39 | 56.52 |

**Note:**
Protein concentration was estimated by Lowry method. The experiments were conducted three times. CA activity was estimated by Wilbur and Anderson method.

*mazatlanica* (Table 4), as described before (Fig. 1), and allowed us to identify it as a N66, a SMPs from the shell of *Pinctada mazatlanica*.

### Recombinant N66 from Pinctada mazatlanica

The recombinant N66 protein was expressed as a soluble form at 37 °C with one mM IPTG in *E. coli* after a 4 h induction, and the bands corresponding to the protein in SDS-PAGE

**Table 4 Native N66 peptide sequences matching with mass spectrometry and deduced amino acid sequences from the N66 gene.**

| Peptide | Calculated mass (Da) | Observed mass (Da) |
|---|---|---|
| QSPINIWSHR | 1,237.38 | 1,237.64 |
| NLHIHIGK | 931.11 | 931.54 |
| HFTPMEAHLVFHH | 1,465.69 | 1,500.73 |
| RWDLENVR | 1,087.20 | 1,087.56 |
| SLDVEITPEMVLPPIK* | 1,781.14 | 1,780.97 |
| RVLDALR* | 842.01 | 842.52 |

Note:
* Peptides obtained only for the 60 kDa band.

analysis agreed with the native N66 of ASM from *Pinctada mazatlanica* (Fig. 3B). The recombinant N66 was successfully purified by affinity chromatography using Ni-Sepharose columns (Fig. S3). The electrophoretic analysis showed two protein bands of ~60 and 64 kDa which agreed with the theoretical molecular weight of the native N66 of ASM from *Pinctada mazatlanica* (Fig. 3B). Recombinant N66 did not show glycosylation after staining with PAS (data not shown).

### Carbonic anhydrase activity of native and recombinant N66 proteins

The CA activity of the native and recombinant N66 was confirmed qualitatively (Fig. 4) and quantitatively. Figure 4 showed comparable yellow halos obtained with the native and recombinant N66, indicating CA activity. Commercial bovine CA was used as positive control. The comparable CA activity of the native and recombinant N66 proteins was confirmed with the Wilbur and Anderson method (1948). The results showed significant differences between the native (161 ± 12.5 U/mg protein) and the recombinant protein (197 ± 7.5 U/mg protein), being the recombinant protein the one with higher activity.

### In vitro crystallization by native and recombinant N66

To elucidate the effect of native and recombinant N66 on the growth of calcium carbonate crystals, three individual in vitro crystallization assays testing the formation of aragonite and calcite were established and analyzed by SEM (Fig. 5). When no protein was added (negative control) to the three salts tested ($NaHCO_3$ + $MgCl_2$, $NaHCO_3$ + $CaCl_2$, and $CaCO_3$), different crystal morphology was observed. The crystals formed were unorganized when sodium bicarbonate was in the presence of magnesium chloride, while in the presence of calcium chloride, crystals become compact and thicker, and with calcium carbonate, the formed crystals exhibited the typical rhombohedral form of calcite (Figs. 5A–5C). In the presence of 0.2 mg protein/mL of native N66 to the three salt preparations, polycrystalline aggregates of aragonite and calcite in presence of $MgCl_2$ and $CaCO_3$ respectively were shown, however in the presence of $CaCl_2$, crystals were small structures of calcite and in a lesser quantity per area (Figs. 5D–5F). In contrast to the native N66, the recombinant form of N66 produced bigger and organized crystal structures plates (Figs. 5G–5I).

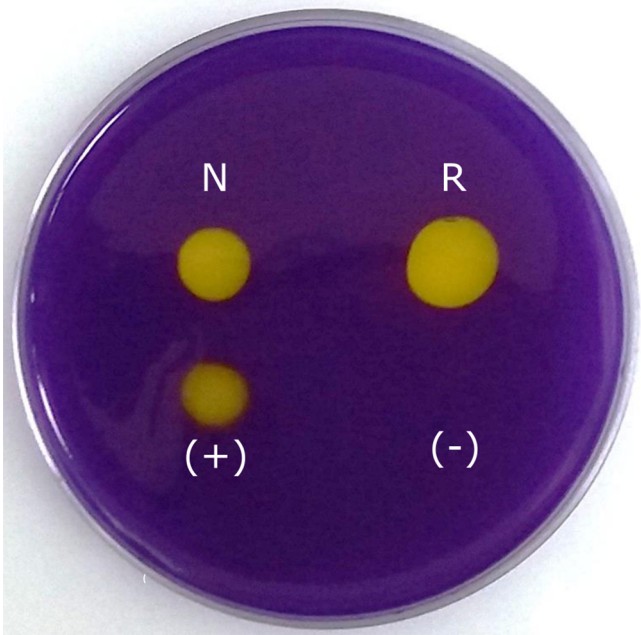

**Figure 4** **Carbonic anhydrase activity of native and recombinant N66 from *P. mazatlanica*.** Carbonic anhydrase activity was visualized by soaking the agarose plate in bromocresol purple, and exposing the plate to saturated $CO_2$ water. Commercial bovine carbonic anhydrase was used as positive control and water as negative control. N: native N66 and R: Recombinant N66 from *P. mazatlanica*.

## DISCUSSION

The mollusk shell is a complex structure made of organic and mineral components. The shells are composed by layers formed of calcium carbonate polymorphs like aragonite or calcite crystals (*Takeuchi et al., 2008*; *Zhang & Zhang, 2006*). Calcium carbonate polymorphs are formed by SMPs, which have been classified as soluble and insoluble based on the two main extraction methods, EDTA and acetic acid (*Marie et al., 2007*; *Kono, Hayashi & Samata, 2000*). Insoluble proteins, are chitin-proteinaceous complexes rich in aliphatic amino acids, such as Gly and Ala, while the soluble proteins have been found to be polyanionic, enriched of Asp; among the soluble proteins, functional enzymes such as CAs have been described (*Kono, Hayashi & Samata, 2000*), which seems contribute to the control of biomineralization.

Most of the information about genes encoding enzymes with CA activity to form nacre in mollusks is about nacreins (*Mann, Edsinger-Gonzalez & Mann, 2012*; *Leggat et al., 2005*; *Miyamoto et al., 1996*). The nacrein gene has been described in *Crassostrea gigas* (*Song et al., 2014*), *Patella vulgata*, *Mytilus californianus*, *Pinctada maxima* and *Pinctada magaritifera* (*Joubert et al., 2010*), while N66 mRNA has only been described in *Pinctada maxima* (*Kono, Hayashi & Samata, 2000*) and *Pinctada margaritifera* (*Joubert et al., 2010*). These homologous genes have been found to be expressed in the mantle (*Miyamoto, Miyoshi & Kohno, 2005*; *Kono, Hayashi & Samata, 2000*), which is a conserved organ involved in shell formation of mollusks. We followed this lead and proved

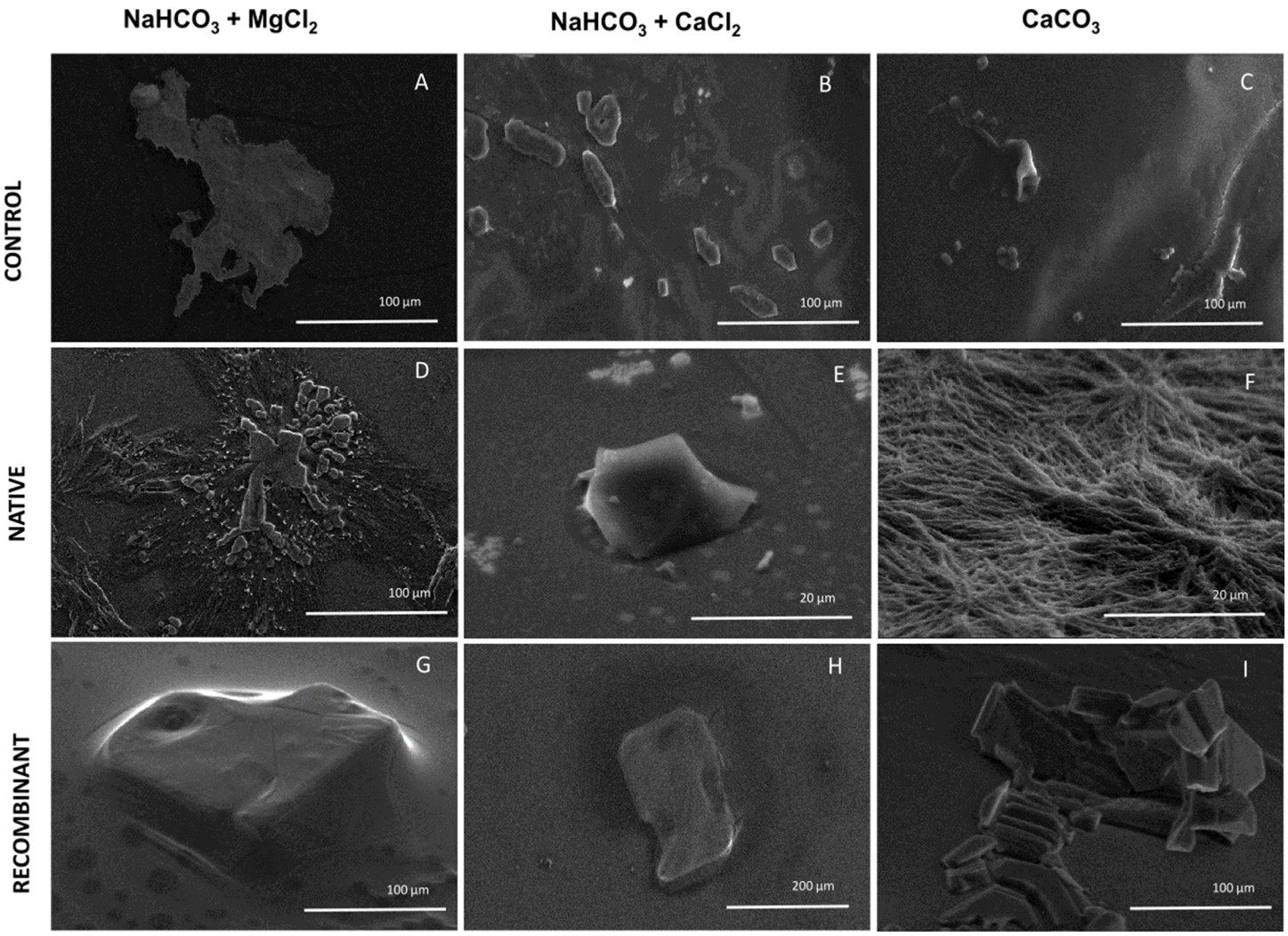

**Figure 5 Effect of native and recombinant N66 from *P. mazatlanica* on CaCO₃ crystal growth.** SEM micrographs of calcium carbonate crystal growth in the presence of 0.2 mg/mL of N66 (native or recombinant) incubated at 4 °C for 21 days. Controls (A–C) Salts without N66; Native N66 (D–F) Salts in the presence of native N66 and, Recombinant (G–I) Salts in the presence of recombinant N66. Pictures are representative of three independent experiments.

that *Pinctada mazatlanica* expresses the N66 mRNA in the mantle and secretes the enzyme into the extrapalleal space to participate in the shell growth.

Only six SMPs with CA activity have been isolated and characterized from pearl oysters, nacreins from the Akoya pearl oyster *Pinctada fucata*, turban shell *Turbo marmoratus*, the edible Iwagaki oyster *C. nippona*, Yesso scallop *Patinopecten yessoensis* and giant clam *Tridacna gigas* (*Norizuki & Samata, 2008*; *Leggat et al., 2005*; *Miyamoto, Yano & Miyashita, 2003*; *Miyamoto et al., 1996*), and only one N66 from *Pinctada maxima* (*Miyamoto et al., 1996*). CAs are usually purified via the agarose-sulfonamide affinity column chromatographic procedure (*Ozensoy, Capasso & Supuran, 2016*), however, in this study a Sepharose-Ni column was successfully used to capture the N66 protein from the shell of *Pinctada mazatlanica*, since it is well known that the CA domain

possesses three conserved histidine residues at the active site and His exhibits the strongest interaction to metal ion matrices, as electron donors groups on the imidazole ring of His form coordinated bonds with the immobilized metal (*Bornhorst & Falke, 2000*). The isolated protein was identified by mass spectrometry as a N66, due to its similarity to N66 from *Pinctada maxima* and its homolog protein, nacrein; native N66 and its recombinant form showed two bands with a molecular weight of ~60 and ~66 kDa, respectively. The double bands of proteins observed on SDS-PAGE under reducing conditions have been also observed in other CAs such as CAH1, CAXII, and CAIX, the last one migrates as a double band between 54/58 kDa under reducing conditions. This effect is most likely due to glycosylations (*Buren et al., 2011*; *Ulmasov et al., 2000*) and/or equivalent populations of mature and pro-forms of the enzyme, both possessing a propensity to become an oligomer in non-reducing conditions (*Hilvo et al., 2008*; *Ulmasov et al., 2000*; *Pastorek et al., 1994*). Since the recombinant N66 do not possess PTMs, and both proteins, native and recombinant, migrate similar on SDS-PAGE, the formation of oligomers of N66 is suggested in this study. However, further experiments are required to establish if the double band of N66 from the shell of *Pinctada sterna* is due to glycosylations or oligomerization of the protein.

Structurally, N66 in *Pinctada mazatlanica* is similar to N66 from *Pinctada maxima* (*Kono, Hayashi & Samata, 2000*) and its homologous proteins, nacreins and N45 (*Miyamoto et al., 1996*), they share in its modular structure a signal peptide, two CA domains and a NG domain. α-CAs contain a domain which is composed by three amino acid residues (Gln, Asn, and Glu) and three His (*Vulo et al., 2008*), according to the deduced primary structure of N66 from *Pinctada mazatlanica*, the protein contains two CA domains composed by $Gln_{125}$, $Asn_{154}$, $Glu_{168}$ and three His residues which act as zinc ligands ($His_{156}$, $His_{158}$, and $His_{181}$), suggesting that N66 possess a functional CA domain. CA activity has been only proven in nacrein from *Pinctada fucata* (*Miyamoto et al., 1996*) and N66 from *Pinctada mazatlanica* and its recombinant protein (this study). The recombinant N66 showed higher CA activity than the native N66 and the bovine CA. The NG domain found in N66 from *Pinctada mazatlanica* differs in length and composition to its homologues, accounting for over ~30% of their sequences. This domain has been proven to function as a negative regulator in nacrein shell formation (*Miyamoto, Miyoshi & Kohno, 2005*), but also as a positive regulator, both as a $Ca^{2+}$ concentrator and as an enzyme required for production of carbonate ions (*Norizuki & Samata, 2008*). It is hypothesized that a long repeat NG domain might have a stronger interaction between the protein with $Ca^{2+}$ ions, matrix components and crystals, leading to a better calcification ability (*Samata et al., 1999*). This suggests, that N66 might have better calcification ability than its homologs, although, further experiments are required to prove this hypothesis.

Post-translational modifications of SMPs have been suggested to play an important role in biomineralization. Phosphorylation and glycosylation are the major players in many protein functions (*Saraswathy & Ramalingam, 2011*). The in silico analysis of N66 from *Pinctada mazatlanica* showed 36 putative sites of phosphorylation and 16 putative sites of glycosylation. The biochemical analysis confirmed that N66 is glycosylated

(Fig. 3B), a modification that has been observed in other matrix proteins (*Montagnani et al., 2011*; *Suzuki et al., 2011*). Glycosylation participates in protein folding and/or interactions with $Ca^{2+}$ ions (*Suzuki et al., 2011*; *Borbas, Wheeler & Sikes, 1991*) and provides sites of high anionic charge through addition of sialic acid or sulfated oligosaccharides (*Michenfelder et al., 2003*) which have been related to promote the uptake of $Ca^{2+}$ ions (*Song et al., 2014*). The saccharide chains of N66 in *Pinctada mazatlanica* probably have similar effects, however, the chemical identity of the oligosaccharides moieties, the number and the location of actual glycosylation sites along the protein of N66 remain to be determined.

Experiments involving in vitro calcium carbonate crystallization have been used to prove functional activity of SMPs since most of the described SMPs do not possess catalytic domains. Crystal formation comprises two main processes, nucleation and growth. It begins with an amorphous calcium carbonate as an intermediate of the crystalline carbonate polymorph such as calcite or aragonite (*Wang et al., 2009*). The presence of $Mg^{2+}$ ions foster a metastable aragonite growth that make up the mineral phase of nacre (*Evans, 2017*; *Su et al., 2016*; *Samata et al., 1999*), and when $Mg^{2+}$ ions are low, they induce the formation of calcite (*Raz, Weiner & Addadi, 2000*). The morphological observation of calcite and aragonite by SEM has been made by several authors, showing that crystals of calcite display a rhombohedral form while aragonite has an orthorhombic form (*Zigovecki, Posilovic & Bermanee, 2009*), although the morphology of each polymorph can be changed due to the crystallization condition, for example, aragonite is needle-like but it changes to flake-like or cauliflower-like (*Chakrabarty & Mahapatra, 1999*). In this study, N66 and its recombinant protein have different CA activity and lead to calcium carbonate precipitation, displaying aragonite and calcite structures when crystallization was made in presence of $MgCl_2$ and $CaCl_2$, respectively. Similar results were obtained by *Kono, Hayashi & Samata (2000)*, for the N66 from *Pinctada maxima*, which induces aragonite layers, however it required the presence of N14 and $Mg^{2+}$ ions. The recombinant N66 fostered a higher crystal growth than the native N66, this evidence raise the possibility that the presence of PTMs (e.g., glycosylations and phosphorylations) affect the growth and shape of calcite and aragonite crystals, which is in agreement to previous evidences in *Unio pictorum*, that showed that glycosylations might change the crystal morphology in in vitro crystallization experiments (*Marie et al., 2007*).

## CONCLUSION

In this study, a N66 mRNA from the mantle of *Pinctada mazatlanica* was identified. The transcript was translated to one single protein named N66. The N66 is a glycoprotein with CA activity and calcium carbonate precipitation ability in vitro, suggesting that N66 is involved in the mineralization process in *Pinctada mazatlanica*. Experiments with the recombinant N66 protein suggest that posttranslational modifications might play an important role in the modulation of the CA activity. The identification of the carbohydrates associated to N66, as well as the determination of other posttranslational modifications (e.g., phosphorylation, sulfation, etc.), and their role in the calcium carbonate precipitation (e.g., crystal growth, type of polymorph) remains to be tested.

Also, understanding the potential oligomerization process of the N66 and its role in the mineralization process remains unclear.

### Funding
The authors received no funding for this work.

### Competing Interests
The authors declare that they have no competing interests.

### Author Contributions
- Crisalejandra Rivera-Perez conceived and designed the experiments, performed the experiments, analyzed the data, prepared figures and/or tables, authored or reviewed drafts of the paper, approved the final draft.
- Catalina Magallanes-Dominguez performed the experiments, approved the final draft.
- Rosa Virginia Dominguez-Beltran performed the experiments, approved the final draft.
- Josafat Jehu Ojeda-Ramirez de Areyano performed the experiments, approved the final draft.
- Norma Y. Hernandez-Saavedra conceived and designed the experiments, contributed reagents/materials/analysis tools, approved the final draft.

### DNA Deposition
The following information was supplied regarding the deposition of DNA sequences:
    The N66 mRNA sequence is accessible at GenBank: MH473230.

### Data Availability
    Evidences of the purification process of the native and recombinant N66 described are available in Figs. S2 and S3.

### Supplemental Information
Supplemental information for this article can be found online at http://dx.doi.org/10.7717/peerj.7212#supplemental-information.

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
