# Peer review of "Biochemical and molecular characterization of N66 from the shell of Pinctada mazatlanica"

_PeerJ, doi:10.7717/peerj.7212_

## Round 0.1 · original submission · Major Revisions

The manuscript deals with the potentially interesting report but its technical aspects are not sufficient to meet standards of PeerJ. However, since two of the three reviewers recommended major revisions and provided the detailed instructions how to improve the paper, I would be glad to see the revised version of the paper addressing ALL the concerns raised by the reviewers. Please answer all the questions point-by-point and change the manuscript accordingly, otherwise justify why specific changes cannot be made. Careful reading by a native speaker is highly recommended, alternatively please use an English language editing service. Please make all efforts to persuade a reader and the reviewers in the identity of the recombinantly produced N66 protein represented by two bands on the SDS-PAGE (see concerns of Reviewer 2) and that it was the glycosylation that inhibited the activity of the native version of the protein isolated from the mollusks and e.g. not other posttranslational modifications (see concerns of Reviewer 3). So far these are the weakest points of the paper. Also, please do take into account that the acceptance of even the extensively revised version is not guaranteed at this stage.

Reviewer 1 ·

Basic reporting

Comments and suggestions:
1. Some grammatical, syntax and spelling errors should be corrected all along the manuscript. English language needs to be improved to be clearer.
Abstract
-The words “homologues” (this word does not exist in the English language) should be replaced by homologs
- Lane 27- and its full-length cDNA sequenced
- Lane 29- two carbonic anhydrase domains
Introduction
- Lane 47- calcite, and nacre), are
- Lane 55- Reference should be corrected (Sanchez)
- Lane 63- C-terminal end
- Lane 65- belong
Methods
- Lane 147- Reference last names should be corrected
- Lane 156- Escherichia coli TOP 10 cells were transformed
- Lane 158- 5 mL of the culture medium were transferred
- Lane 161- incubated. Bacterial cells
- Lane 166- Recombinant His-tagged proteins were purified
-Lane 196- As described by Laemmli (1970) and stained with silver nitrate in 10% acrylamide gels. To prepare…
- Lane 211- Native or recombinant??
- Lane 220- 10 mL of 0.2 mg protein/ mL
-Lane 232- 5 mL of 0.012 M
-Lane 234- magnetic stirring bar
-Lane 335- were added
-Lane 247- As a control,
-Lane 248- Each solution experiment was made in triplicates and incubated
-Lanes 280-281- O-glycosylation and phosphorylation sites are…
-Lanes 291 – and silver stained gel were observed
-Figure 3 caption- coomassie
-Figure 4 caption- exposing the plate to saturated…
-Lane 324- in the presence of 0.2 mg protein/ mL
Discussion
-Lane 373- its recombinant form/protein
-Lane 381- this suggests
-Lane 389- glycosylation plays
-Lane 408- its recombinant protein

Experimental design

2. Methods
a) Some sample data are missing. How many organisms were sampled?
Do samples come from one single adult/larvae? Male/female? Or from a pool?
b) Authors should mention in this section how/why they suggest the native N66 protein was His-tagged to be then isolated by affinity chromatography? It is not clear how do they know these proteins had a His-residues tag exposed to be correctly attached to the Nickel column? Did they build a predictive model of the protein? If they did it, it would be really helpful to include it in the manuscript to reinforce their findings and explanations.
c) After proteins purification, it is not clear whether authors are talking about native or recombinant proteins in the SDS-PAGE and mass spectrometry analysis, it should be clearer.

Validity of the findings

3. Results
a) Lane 262- Data was not confirmed since no record was found at NCBI.
b) Table data are not required.
c) Figure 2 should be replaced by a multiple alignment including the same protein sequences that were drawn in figure 2.
d) Table 4 data are already included in the results section, the table is not needed.
e) Figure 3. Authors should clearly explain why two bands were observed? Are there two N66 proteins in the organism extract? Why are there two proteins from the recombinant extracts?
As I understand two proteins may occur naturally in native extracts as isoforms probably, but what about recombinants?
f) If native N66 proteins are O-glycosylated, it would be expected that their molecular masses were higher than those of recombinant proteins which are supposed to be no-glycosylated or phosphorylated… Gels proteins do not match with described results since those from the periodic acid/Schiff staining should be larger and no smaller as the figure shows.
4. Discussion
a) What does it means that O-glycosylation and phosphorylation sites are flanking the NG domain? Why is glycosylation required for these proteins?
b) A clear explanation is expected to be given about the two bands, and the no differences in the molecular masses of the native and recombinant forms of N66.
5. Conclusion
As part of the conclusion section, it would be desirable for authors to state clearly their main findings and future perspectives.

Additional comments

Manuscript ID: 35907 Peer J
Title: Biochemical and molecular characterization of N66 from the shell of Pinctada mazatlanica
Authors: Crisalejandra Rivera-Perez, Catalina Magallanes-Dominguez, Rosa Virginia Dominguez-Beltran, Xehu Ojeda- Ramirez, Norma Yolanda Hernandez-Saavedra

The manuscript is about the shell soluble protein N66 from P. mazatlanica. As only a few similar proteins have been studied in other mollusks species, results contribute to gain original knowledge about the mechanisms that mollusk have developed to maintain their integrity and survive. Some major and minor aspects were found to improve the manuscript before it can be accepted for publication.

Reviewer 2 ·

Basic reporting

no comments

Experimental design

no comments

Validity of the findings

no comments

Additional comments

This paper deals with characterization of N66 isolated from the shell of P.mazatlanica and also its recombinant protein expressed in E.coli.
Many shell matrix proteins have been isolated from nacreous layer of bivalve and N66 is one of them. So, it is not much interesting if papers are merely devoted to similar results reported before. Although the reviewer could not understand well this paper due to poor English writing, it seems that this paper does not provide any new findings compared with those reported previously.
Furthermore, there are some mysterious results for examples as described below.
1) SDS-PAGE analyses show intensities of protein bands not dependent on concentrations of various fractions obtained by chromatography (Fig, S2 and S3). This raises the possibility that these are proteins contaminated during purification process.
2) Recombinant protein from E.coli repression system should give only one protein without any isoforms because this system give no post-translation modification. However, this paper gives two protein bands in SDS-PAGE (Fig.3).
3) No spectral data for LC/MS are given in this paper, which doubts any guarantee for N66 obtained in this paper.
4) Other ambiguities are also found in cDNA cloning, Fig. 4 and Fig. 5.

Reviewer 3 ·

Basic reporting

It is well known that post-translational modification plays a very important role during the protein interactions, as well as in the process of biomineralization. The author mentioned that a higher carbonic anhydrase and crystallization ability has been observed in in vitro prokaryotic expression of the recombinant N66, rather than the natural N66 extracted from oyster shell, indicating that the modification of the N66 protein might inhibit its own functions. Although, it‘s meaningful for us to understand how matrix proteins regulate shell formation, the overall conclusion is not well supported by the authors’ experimental findings. The experiments can be further improved.

Experimental design

The experiments can be further improved.
a. Compare carbonic anhydrase and crystallization ability of recombinant N66 with native one, after in vitro glycosylation. And also compare both activitis of the native N66 with recombinant one, after de-glycosylation.
b. Does Native N66 have other post-translational modifications? If so, can it also affect carbonic anhydrase and crystallization ability of N66?

Validity of the findings

The conclusion is not accurate.
The author made a conclusion that the lack of posttranslational modifications in the recombinant N66 modulates its activity via simply comparing the carbonic anhydrase and crystallization ability of natural N66 with recombinant N66. However, protein post-translational modifications are not only glycosylation modifications, but also other post-translational modifications such as phosphorylation.

Additional comments

no comment

---

## Round 0.2 · Major Revisions

Although both reviewers suggest now to accept this paper, I cannot do this until the situation with the double bands in fractions of the recombinant version of N66 is completely clarified by the authors. The previous response is not satisfactory and there is still a high risk that the two bands seen in all gels correspond to an abundant contaminant like keratin (hands, hair, etc.). This is indirectly supported by the fact that there is no difference in the intensity of those bands nor their ratio during the purification process (where enrichment with the protein of interest is expected instead!). Would those two bands correspond to monomers and dimers, as authors suggest, they would differ by mass twofold (66 and 132 kDa), not to mention higher-order oligomeric species, which will give much even higher masses, tetramers will give 264 kDa. Such -S-S- stabilized dimerization indeed takes place in some proteins, even in the presence of reducing agents, but the mass difference cannot be as we see in the case reported in the manuscript. Densitometry of the FIgs S2-3 is done very poorly and is not convincing as, basically, gels do not correspond to the peaks on the plot. For example, E5 intensity on the gel in Fig. S3 is not lower than that of E3, but the plot does show this difference. An explanation that the two bands may be a result of PTM is not acceptable as well because the recombinant protein gives the same pattern as the native one whereas PTMs are not expected in E.coli. In addition, the 1D MS data of a few peptides (new Table 4) cannot serve as solid confirmation of the identity of a protein and MS/MS sequencing of selected peptides should have been done to really confirm the identity of N66. Also, it is said LC/MS, but was it done by excising of the bands from SDS-PAGE gel and subjected them to digestion and MS, or the “purified” sample was directly analyzed by LC/MS? The first case is required, also to assign protein bands to a specific set of peptides. Please include further in-depth analysis in the manuscript to convince a reader.

Reviewer 1 ·

Basic reporting

I deeply reviewed the last version of the manuscript. All comments were taken by authors, and mistakes were corrected. The corrected English version is clear and fully understandable.

Experimental design

Each of the major concerns about methods and missing date were corrected. This section is now described in sufficient detail and clarity.

Validity of the findings

New and original data are part of the manuscript, perspectives were also included. All data were validated and confirmed to be correct from the newly added explanations.

Additional comments

THE MANUSCRIPT PROVIDES BASIC KNOWLEDGE ABOUT THE SHELL SOLUBLE PROTEIN N66 FROM P. mazatlanica. INFORMATION IS NOW COMPLETE AND CLEAR. RESULTS WERE CORRECTLY ANALYZED AND COMPARED WITH OTHER PREVIOUS STUDIES TO BETTER UNDERSTAND THE PARTICULARITIES OF THIS ORGANISM PROTEIN. THE MANUSCRIPT SHOULD BE NOW ACCEPTED.

Reviewer 3 ·

Basic reporting

no comment

Experimental design

no comment

Validity of the findings

no comment

Additional comments

The mechanism on the regulatory role of matrix proteins during biomineralization in mollusks remains unclear. It's great that the auther used the modifications of matrix protein as a springboard for studying its functions. I hope the auther will dig more deeper.

---

## Round 0.3 · accepted · Accept

The response 2 is only partly satisfactory, again. However, I'm now accepting your paper under your own responsibility. Please double check grammar, several inaccuracies are already seen in the abstract.

#